# DISTRIBUTION AWARE METRICS FOR CONDITIONAL NATURAL LANGUAGE GENERATION

## ABSTRACT

Traditional automated metrics for evaluating conditional natural language generation use pairwise comparisons between a single generated text and the best-matching gold-standard ground truth text. When multiple ground truths are available, scores are aggregated using an average or max operation across references. While this approach works well when diversity in the ground truth data (i.e. dispersion of the distribution of conditional texts) can be ascribed to noise, such as in automated speech recognition, it does not allow for robust evaluation in the case where diversity in the ground truths represents signal for the model. In this work we argue that existing metrics are not appropriate for domains such as visual description or summarization where ground truths are semantically diverse, and where the diversity in those captions captures useful additional information about the context. We propose a novel paradigm for multi-candidate evaluation of conditional language generation models, and a new family of meta-metrics built on top of existing pairwise metrics that compare the *distributions* of reference and model-generated caption sets using small sample sets of each. We demonstrate the utility of our approach with a case study in visual description: where we show that existing models optimize for single-description quality over diversity, and gain some insights into how sampling methods and temperature impact description quality and diversity.

## 1 INTRODUCTION

Recent models for conditional language generation, particularly in the field of visual description, have shown dramatic improvements in both fluency and the ability to ground generated language in context (Liu et al., 2021; Zhou et al., 2020; Mokady et al., 2021; Chen et al., 2018). Standard metrics for these tasks such as BLEU, ROUGE, METEOR, and CIDEr, compare a generated text with a reference set of texts and compute some measure of quality for the generated text. By construction of these metrics, a model will achieve the best performance by generating a single high-scoring text. In contrast, it has been widely observed that large language models such as GPT-3 (Brown et al., 2020) or LAMDA (Thoppilan et al., 2022) generate the most realistic texts at temperatures close to one, where the set of potential texts generated is often very diverse. More significantly, if we look at an example of an image from MS-COCO and its set of reference captions (Figure 1), we notice that each (human-generated) reference contains a unique subset of the overall information in the image:

"A woman in a red robe is sitting at a dining table."
"A woman in a red flowered shawl sits at a table while a man wearing jeans is in the kitchen looking at her."
"A person sits at a table and another person stands in the kitchen."
"A woman is sitting at a table wearing a robe while a man is cooking."
"Man and woman in a kitchen looking in the same direction."

Important features like the red robe, the man, the gaze of the two people etc, are mentioned only in one or a few captions. Metrics that encourage generating information from *only one* of these captions will generally fail to capture much of the important detail in the image. This holds for more than just image description. For many conditional language generation tasks such as video captioning, abstractive summarization, translation, and open-ended question-answering, it is often beneficial to be able to sample from a diverse distribution of generated outputs.

If we compute a caption from a state-of-the-art model (Zhou et al., 2020) we get:

"A woman sitting in a kitchen next to a man."

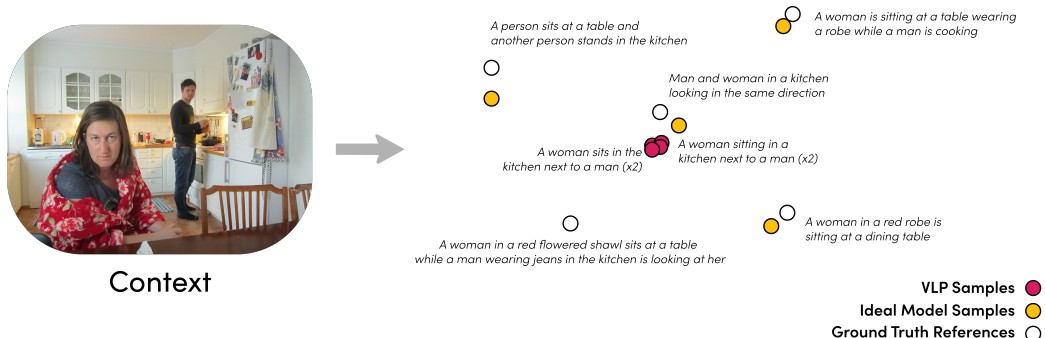

Figure 1: Samples from these two models achieve similar BLEU scores, however, the samples from VLP lie near a center of the distribution, and fail to capture the dispersion of natural language in the ground truths, while the samples from an ideal model better match the ground truth distribution. In this work, we introduce metrics which better measure deviations between samples from candidate and reference distributions, compared to single-sample pairwise metrics.

In this description, we see that only information common to most or all of the reference captions is preserved. This is intuitive, since including more information runs the risk that no reference caption contains that information, leading to a low score. Note that this caption may not actually be the most likely (highest expected similarity to a reference caption), because e.g. the BLEU score also includes a term encouraging longer texts. It seems the designers of these metrics are already aware that direct use of shortest distance to a reference caption favors generated captions which are even shorter and more impoverished. However, the (log-) text length heuristic in standard metrics is a poor proxy for actual diversity. Instead of generating a variety of captions, taking 10 samples from the state-of-the-art model, yields only 10 repetitions of the above caption.

Thus, we encounter an issue in the evaluation of conditional text generation models from multiple sampled texts. When several ground truths are available, typically the metric score is based on the maximum score with some ground truth. This leads to an issue, shown earlier, where the model is encouraged to produce a text with the lowest expected distance (max pairwise score for a particular $n$-gram as in BLEU) to a reference text, i.e. near a strong mode in the training text distribution. Changing the metric aggregation method, say from max score over reference examples to average or sum (ROUGE), does not change the situation substantially. The model will still be encouraged to produce a single output with high average scores to nearby references, which will be maximized at a smoothed mode in the training text distribution. Failure modes of other methods of aggregation are discussed in both Caglayan et al. (2020) and Yeh et al. (2021), including issues with multi-modal reference distributions and single outlier texts.

Such an over-reliance on simple aggregations for multiple candidates and references has, over time, compounded into several issues: The first, discussed further in section 3, is that, as observed in visual description by Chan et al. (2022) and dialog generation by Caglayan et al. (2020), human performance on datasets under existing metrics is often lower than model performance, even though human-generated captions tend to receive higher scores under human evaluation. The second, discussed in section 2, is that diversity of candidate texts is largely relegated to reference-unaware measures, encouraging models to diverge from ground truth distributions to hit diversity targets.

In this work, we aim to solve these problems by introducing several novel automated ways of measuring the performance of conditional text generation models based on measuring the divergence between samples from two text distributions. While some recent methods have been designed to closely measure the divergence between full distributions of text data in the unconditional case (Pillutla et al., 2021), no such methods exist for the conditional generation case, which operates on the level of 10s of reference samples and candidates. Our contributions are summarized as follows:

1. We introduce a new paradigm for the evaluation of conditional text generation models based on sampling from both candidate and reference distributions.

2. We introduce two new families of metrics which *extend* existing semantic distances: triangle-rank metrics, and kernel-based metrics, designed to measure the divergence between small text samples from candidate and reference distributions.

3. We explore how our new metrics behave in the context of visual description (both image and video description) and show that by measuring distributional effects, we can capture nuances in the data that existing metrics cannot explore.

## 2 RELATED WORK

This work is not the first to notice the shortcomings of traditional metrics for the automated evaluation of conditional language generation models. In visual dialog, Caglayan et al. (2020) find that a number of the automated metrics proposed for visual dialog do not match well with human judgment, while in visual description, Chan et al. (2022) find that current automated metrics do not assign high scores to human-generated descriptions. This work not only quantifies such issues but proposes a method for addressing these cases without developing novel metrics for measuring text semantic distance. In this section, we review related works, roughly divided into three groups; methods for evaluating text quality, text diversity and distribution aware metrics.

**Measuring the Quality of Generated Text:** The evaluation of machine-generated text has long been an active area of research, which has continuously evolved to keep pace with accelerating advances in text generation. As a consequence of the tools available and the state of early text generation approaches, classical measures have primarily focused on evaluating the quality of generated text with respect to ground truth references using surface-level text statistics. Most notably, these include $n$-gram matching based metrics like BLEU (Papineni et al., 2002), METEOR (Banerjee & Lavie, 2005), ROUGE (Lin, 2004), and CIDEr (Vedantam et al., 2015). More recently, the rapid progress enabled by large-scale language models has motivated new evaluation techniques which go beyond superficial n-gram statistics and toward measures that aim to capture the underlying semantics of language (Shimanaka et al., 2018; Clark et al., 2019; Zhang* et al., 2020; Sellam et al., 2020). These approaches leverage high dimensional representations of generated and reference text provided by a state-of-the-art language model, such as BERT (Devlin et al., 2018) in the case of BERTScore (Zhang* et al., 2020) and BLEURT (Sellam et al., 2020). While such methods are focused on measuring the semantic distance between two pairs of natural language texts, the evaluation of the diversity of the generated captions has largely been done independently of quality.

**Measuring the Diversity of Generated Text:** Until recently, measures of diversity for generated text have been largely secondary to measures of quality, since the pursuit of human-like generated text has been the primary focus of the field. In fact, many diversity measures quantify surface-level statistics of the generated text (Van Miltenburg et al., 2018), such as metrics based on the number of unique tokens, unique sentences, or unigram frequency statistics, such as Zipf coefficients (Holtzman et al., 2019). Similarly, $n$-gram-based diversity measures such as self-BLEU (Zhu et al., 2018), compute scores between samples from a model. Unfortunately, these approaches do not consider the diversity of a model's outputs with respect to the diversity of human references. Such measures are also primarily focused on the diversity of the vocabulary, rather than the aggregate semantic diversity, a factor that our proposed work aims to address.

**Distribution Aware Measures of Generated Text:** Recently, MAUVE, proposed by Pillutla et al. (2021), addresses the single-text evaluation paradigm by measuring the divergence between multi-candidate samples and multiple ground truths using density estimates in a text embedding space. Such an approach has the power to measure both dispersion of the text and the quality of the generated text simultaneously. In this respect, MAUVE is the most similar work to ours, in that it proposed a distribution-aware metric. That being said, MAUVE is designed for the setting of unconditional text generation, where many ground truth samples are available, and the entire dataset of human reference texts is compared to model outputs. Unfortunately, while MAUVE works well in these scenarios, it does not work well when only a few references are available (due to its K-means distributional approximation) (see appendix B.5). Such a low-reference scenario is common in conditional NLG, making MAUVE unsuitable for many potential applications.

## 3 METHODS

In this section, we introduce our two primary contributions. First, we introduce and demonstrate the need for a paradigm for multiple candidate evaluation for conditional language generation, and

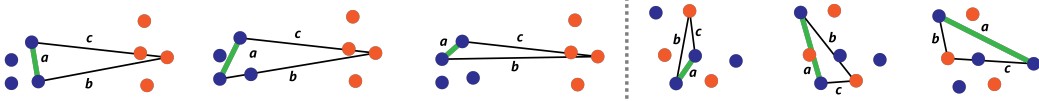

Figure 2: Intuition for TRMs. For samples from different distributions (left), in-distribution edges will often be short, but for identical distributions (right), edge rank-distributions will be more uniform.

second, we introduce several simple augmentations to existing pairwise metrics, designed to alleviate the sensitivity issues induced by evaluating conditional language generation models with only a single candidate text. Our family of augmented metrics, which we dub Triangle-Rank Metrics (TRMs), represents the first step towards optimizing metrics that force models not only to generate samples at the locus of a distribution but also with sufficient variance, hopefully alleviating the field-wide issues that optimizing standard pairwise-metrics can induce.

## 3.1 MULTIPLE CANDIDATE/MULTIPLE REFERENCE EVALUATION

Traditionally, most methods for conditional language generation have been designed to sample a single candidate example using beam search, designed to be a maximum likelihood sample of the data. This single candidate is compared against the reference data. Unfortunately, as discussed in section 1, models can easily exploit such aggregations. For example, when the best score amongst the ground truths is chosen (the "min-distance" aggregate), models generate texts optimizing the *expected minimum distance to the reference distribution*. Such a text is, by definition, the mode of the distribution. While we don't know exactly what this looks like in the case of natural language, the mode likely represents some amount of central tendency. In practice, we observe such central captions tend to be bland and uninformative, as demonstrated in Chan et al. (2022) for visual description, Yang et al. (2019) for image generation, and appendix B.3.

Thus, a single candidate may not be sufficient to understand if the model has learned to approximate the reference distribution. Consequently, we aim to develop methods that can sample several suitable candidate texts, each with high accuracy, while matching the diversity of the ground truth distribution. In this work, to extend methods to multiple candidate generation, we leverage temperature-based sampling or nucleus sampling (as indicated) to produce multiple candidates from each model's distribution. While beam search can generate multiple candidates, Vijayakumar et al. (2016) showed diversity among beams is relatively poor, leading to samples that diverge from the model distribution. This gives us a model which *generates multiple candidate samples*, and requires an evaluation metric which *compares multiple candidate samples to multiple reference samples*.

Note that a sampling approach to generating diverse captions does not preclude a future effort toward generating single "omnibus" captions, which capture detail from many diverse captions. However, such captions will be much longer than typical human captions, and will score poorly under the standard metrics, as they would be quite different from (much more detailed than) individual reference captions. A path toward such omnibus captions (assuming they are practically useful), would be to first generate diverse human-like captions, and then summarize a set of them into a single text. The present work still supports the first step of diverse caption set generation and optimization.

**Extending Existing Metrics for Multi-Candidate Evaluation** Currently, no standard pairwise metrics (BLEU, METEOR, CIDEr, ROUGE, CIDEr, SPICE) support a comparison between multiple candidates and multiple references, and the most efficient extension of existing metrics to multi-candidate, multi-reference situations is a non-trivial task. In this work, we naively extend the existing pairwise metrics (Papineni et al., 2002; Agarwal & Lavie, 2008; Lin, 2004; Vedantam et al., 2015; Zhang* et al., 2020) to multiple candidates through the use of mean aggregation. Thus, for a standard pairwise score $S$, set of candidates $(c_1, \ldots, c_n) = C$ and a set of references $(r_1, \ldots, r_m) = R$, we assign the output score $S_{\text{agg}}$ as:

$$S_{\text{agg}} = \frac{1}{N} \sum_{i=1}^{N} S(c_i, R) \tag{1}$$

## 3.2 TRIANGLE-RANK METRICS (TRMS)

Existing metrics for semantic similarity are extremely powerful for determining pairwise semantic distances between two utterances (Papineni et al., 2002; Agarwal & Lavie, 2008; Lin, 2004; Vedantam

et al., 2015; Anderson et al., 2016), thus, it makes little sense to discard the existing research when extending metrics to a multiple-candidate evaluation. Further, these metrics are already well correlated with human judgements of quality - thus, we need only to increase the sensitivity of these measures when faced with multiple candidates and references, rather than rebuild the entire set of distributional measures. How, then, can we leverage already strong pairwise tools in a multiple candidate scenario? Unfortunately, many statistical techniques for measuring the distances between samples require points to lie in a metric space (Basseville, 2013) - however, most text distances neither respect symmetry nor triangle inequality.

We propose an answer based on an application of the triangle-rank statistic for statistical testing proposed by Liu & Modarres (2011). The triangle-rank statistic has several promising properties: it neither requires symmetry nor the triangle inequality in the metric space (it only requires $d(x, x) = 0$), and it is computed using only pairwise distances, meaning that we can easily reuse existing text semantic distance functions when computing the statistic.

For the purpose of explanation, it can be helpful to think of texts as points on an arbitrary manifold (based on the selected text distance function). To compute the triangle-rank statistic for a given distance $S$, a set of candidates $(c_1, \ldots, c_n) = C$ and a set of references $(r_1, \ldots, r_m) = R$, we first extract all directed triangles $(t_1, \ldots) = T$, such that one point lies in $C$ and two points lie in $R$. We refer to the edge between points from the same distribution as $e_{t_i}^{\text{IN}}$ and the other two edges as $e_{t_i}^{E_0}$ and $e_{t_i}^{E_1}$. We then compute the score for each of the edges. For $(a, b) = e_{t_i}^{\cdots}$, let

$$d(e_{t_i}^{\cdots}) = S(a, b) \qquad (2)$$

We then compute indicators $I_0, I_1, I_2$ for each triangle $t_i$ as follows:

$$
\begin{aligned}
I_0(t_i) =& 1 \text{ if } d(e_{t_i}^{\text{IN}}) \leq d(e_{t_i}^{E_0}), d(e_{t_i}^{E_1}) \text{ else } 0 \\
I_1(t_i) =& 1 \text{ if } d(e_{t_i}^{E_0}) \leq d(e_{t_i}^{\text{IN}}) \leq d(e_{t_i}^{E_1}) \text{ or } d(e_{t_i}^{E_1}) \leq d(e_{t_i}^{\text{IN}}) \leq d(e_{t_i}^{E_0}) \text{ else } 0 \\
I_2(t_i) =& 1 \text{ if } d(e_{t_i}^{E_0}), d(e_{t_i}^{E_1}) \leq d(e_{t_i}^{\text{IN}}) \text{ else } 0
\end{aligned}
\qquad (3)
$$

These indicators represent the rank of the same-sample edge (if it is the smallest, largest, or middle-sized edge). The directed statistic for the sample $(C, R)$, $Q(C, R)$ is then computed as:

$$Q(C, R) = \left| \frac{\sum_{t_i \in T} I_0(t_i)}{|T|} - \frac{1}{3} \right| + \left| \frac{\sum_{t_i \in T} I_1(t_i)}{|T|} - \frac{1}{3} \right| + \left| \frac{\sum_{t_i \in T} I_2(t_i)}{|T|} - \frac{1}{3} \right| \qquad (4)$$

For the experiments in this paper, we use an extension of the directed statistic, the undirected statistic, $TRM(C, R) = Q(C, R) + Q(R, C)$, which increases the sensitivity of the metric by taking into account rank statistics of both within-candidate and within-reference edges.

An intuition for how this statistic measures divergence between distributions is given in Figure 2. If the in-distribution edges are always short compared to the cross-distribution edges, this suggests that either the distance between the candidate and reference distributions is high (different locus), or the spread of the candidates in the semantic space is significantly less than that of the references (different spread). If the in-distribution edge is always the longest edge, it suggests that the spread or dispersion of the candidate samples is higher than the dispersion of the reference samples. Because this statistic takes into account the full distribution through triplets of samples, it does not suffer from the issues with aggregation discussed in section 1 and earlier in this section. Not only does it solve these issues, the TRM extensions build on the existing pairwise metrics, allowing us to increase the sensitivity of the metrics while retaining the existing semantic distance measure and intuitions.

### 3.3 KERNEL-BASED METRICS

While TRMs represent one method of augmenting existing pairwise metrics, a second possible approach relies on representing utterances as points in the embedding space of a model, particularly a large pre-trained model such as BERT (Devlin et al., 2018) or GPT (Brown et al., 2020). Evaluating the distance between two distributions based on representative samples on a Euclidean manifold is relatively well studied in GAN literature. One option, MAUVE, introduced by Pillutla et al. (2021), uses a K-Means density estimator to estimate the distribution of the points on this manifold and then computes a fixed divergence (such as Kullbeck-Libeller) between the two density estimates. Unfortunately, MAUVE can struggle to correctly estimate the density when there are few samples, such as in the case of conditional language generation, as the K-means density estimator is poorly

Table 1: The p-value for the test dataset (using single-video tests, aggregated using HMP (Wilson, 2019) for tractability) generated using standard metrics under the current method of predicting a single best text sample. With a single candidate text (the current evaluation paradigm), the metrics are unable to make a statistically significant distinction between ground truth samples. Additional experimental detail in appendix A.5.

| Dataset | Model | BERT | CIDEr-D | BLEU@4 | METEOR | ROUGE-L |
|---------|-------|------|---------|--------|--------|---------|
| MSR-VTT | TVT (Chen et al., 2018) | 0.6582 | 0.4098 | 0.7813 | 0.4573 | 0.4771 |
|         | O2NA (Liu et al., 2021) | 0.6455 | 0.4574 | 0.7959 | 0.5646 | 0.5938 |
|         | Human | 0.5153 | 0.5314 | 0.8291 | 0.5306 | 0.5669 |
| MS-COCO | CLIPCap (Mokady et al., 2021) | 0.5581 | 0.8223 | 0.8788 | 0.7483 | 0.7985 |
|         | VLP (Zhou et al., 2020) | 0.5925 | 0.7424 | 0.8593 | 0.6644 | 0.7706 |
|         | Human | 0.6406 | 0.6686 | 0.8743 | 0.6358 | 0.6841 |

suited to such situations. Several possible extensions to MAUVE could be considered as an alternative family of distribution-aware metrics, which we dub "Kernel-Based Metrics" (KBMs).

**Frechet BERT Distance**   The Frechet Inception Distance (Salimans et al., 2016) represents the squared Wasserstein distance between multidimensional Gaussian distributions fitted to the components of the input. In the Frechet BERT Distance metric, we replace the Inception embeddings with those from a pre-trained BERT model (Devlin et al., 2018). See appendix A.6 for details.

**MMD-BERT**   A related metric is the maximum mean discrepancy distance function (Li et al., 2017), which leverages a density estimate of the data, and computes the maximum mean discrepancy between the density estimates for each sample. In our case, we leverage a Gaussian kernel estimate over the embeddings generated by a pre-trained BERT model (Devlin et al., 2018). See appendix A.7 for details. While we primarily explore BERT-based embeddings, we explore additional embedding methods in subsection B.1.

## 4   CASE STUDY: VISUAL DESCRIPTION

Visual description is a challenging conditional natural language task, which requires that a model produce a natural language description of a visual context containing objects, actions, and relationships present in a scene. Data sets for visual description often set themselves apart from other datasets for conditional natural language generation (such as those for translation and summarization), as they contain more than one ground truth sample, making it possible to evaluate visual description models using reference data that has already been collected. In this set of experiments, which look at two datasets for visual description, MSCOCO (image description) (Lin et al., 2014) and MSR-VTT (Xu et al., 2016) (video-description) (full dataset details in appendix A.2), we demonstrate that current metrics are not sensitive enough to evaluate the performance of several existing models with our new metrics. We then demonstrate quantitatively how a multi-candidate evaluation paradigm can close this gap, and how a more sensitive metric, such as TRMs or KBMs, can provide new model insights.

**Existing single-ground truth comparison is not sensitive enough**   A natural first question to ask when evaluating the performance of a metric is, "given the existing data, is the metric sensitive enough to distinguish between a model and a reference distribution?" To answer this question, we evaluated the p-values for several existing metrics on a single video. The results, shown in Table 1 demonstrate that using a single description for the candidate dataset is insufficient to tell even known different distributions apart, motivating a transition to a paradigm with significantly more sensitivity. This is a similar result to the observations made in Yeh et al. (2021) and Liu et al. (2016) for dialog generation: most metrics are unable to produce significant results using existing techniques. Thus, even for standard metrics, it makes sense to sample more than one ideal candidate description and aggregate the metric score across these candidate descriptions.

**TRM and KBM metrics are more sensitive than naive aggregation**   In section 3, we proposed several new metrics which can be leveraged by switching to multi-candidate evaluation. Figure 3 shows the sensitivity of both the newly introduced metrics and existing metrics using the naive aggregation schemes discussed in section 3, as we increase the number of candidate samples from the model. While the sensitivity increases for all models to significance, our proposed metrics are much more sensitive with fewer candidate and reference descriptions. As an additional check, when tested

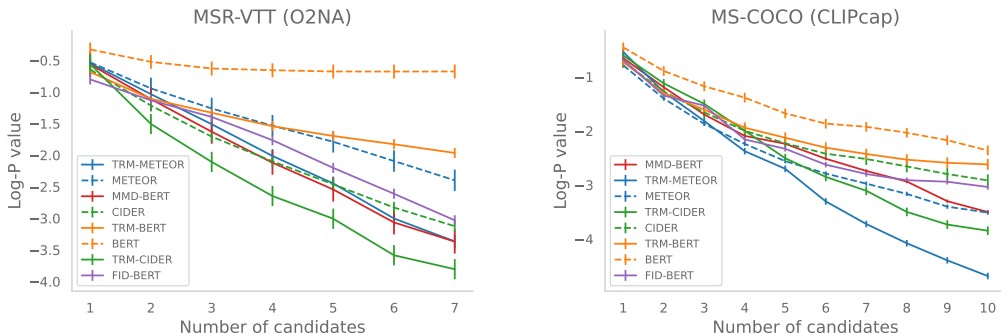

Figure 3: Plots showing the log p-values for the existing and proposed metrics as we increase the number of sampled candidate descriptions from the models. $\text{TRM}_{\text{METEOR}}$ achieves a 162% increase in sensitivity over METEOR, while $\text{TRM}_{\text{CIDEr}}$ represents a 49.3% increase over CIDEr-D for O2NA evaluated on the MSR-VTT dataset. Additional experimental details are given in appendix A.5.

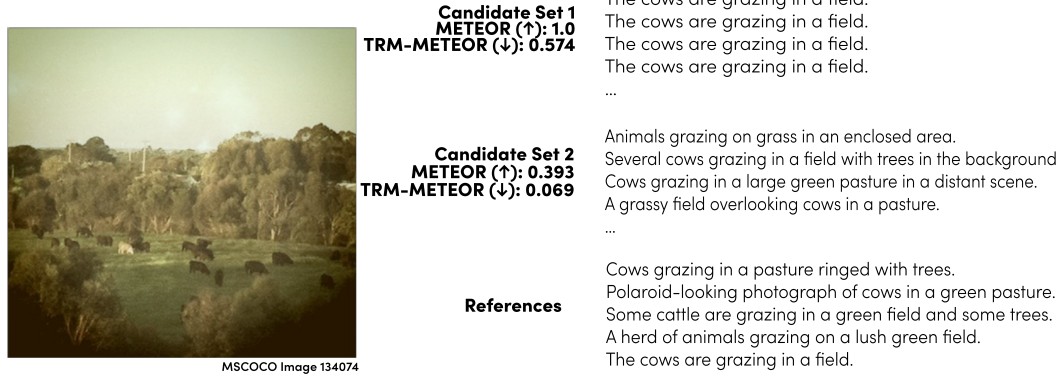

Figure 4: A qualitative sample from CLIPcap. Candidate set one uses beam search (8 beams), while candidate set two uses nucleus sampling (with temperature one, top-k of 20 and top-p of $0.9$). As the diversity increases, the $\text{TRM}_{\text{METEOR}}$ divergence decreases, but METEOR fails to correctly capture the diversity/correctness trade-off, leading to decreased scores for more complete caption sets that are still relatively high quality. Additional qualitative examples are provided in appendix B.6.

on human captions, our metrics do not consider the two distributions significantly different ($p > 0.05$, see appendix B.4). Our proposed metrics do not alter the manifold: so, for example, $\text{TRM}_{\text{METEOR}}$ and METEOR measure the same underlying intuitive divergences (n-gram recall with some additional synonym matching), however, our TRM method increases the sensitivity of the test, allowing us to measure the full distribution divergence, instead of using naive aggregates. It is useful to note that when computing the metrics for a practitioner, computing the p-value of the data is unnecessary (we need only sample enough candidates so we can be sure of the statistical significance of the metric).

**Multi-candidate evaluation illustrates a diversity vs. likelihood trade-off**   A metric's sensitivity to the full distribution has the power to give us novel insights into the visual description task. Consider the two models, VLP (Zhou et al., 2020), a standard transformer-based model pre-trained on large-scale vision and language data, and CLIPCap (Mokady et al., 2021), a transformer-based model which is initialized with a large language model, and uses prefix-tuning with CLIP (Radford et al., 2021a) embeddings (Additional details in appendix A.3). While VLP (Zhou et al., 2020) outperforms CLIPcap (Mokady et al., 2021) in the standard scoring methods, CLIPcap outperforms VLP in the $\text{TRM}_{\text{METEOR}}$ metric. Why is there such an inversion? We can begin to understand the results when looking a bit closer at Figure 5, which plots sampling temperature from the model, against the performance on the $\text{TRM}_{\text{METEOR}}$ and standard METEOR metrics. At low temperatures, where the model is sampling from single maximum-likelihood estimates, VLP outperforms the CLIPcap method. As the temperature increases, there is an inversion in the model performance, and at higher temperatures, CLIPcap outperforms VLP. On the other hand, in the standard METEOR metric, as we

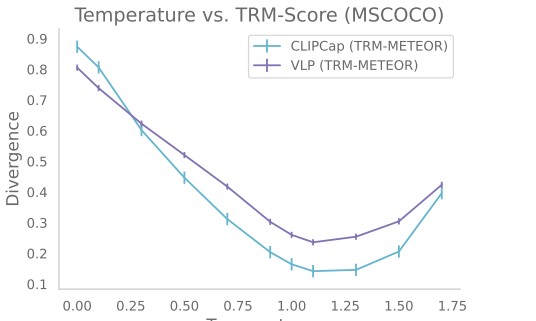 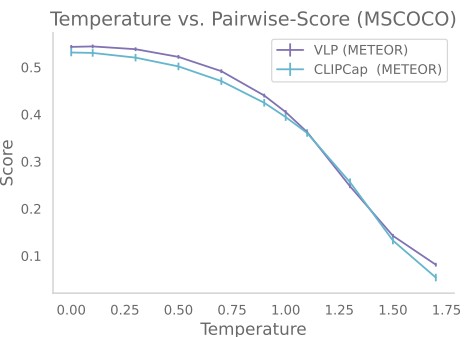

Figure 5: Plots indicating the impact of temperature on the metric scores. Left: $\text{TRM}_{\text{METEOR}}$ ($\downarrow$) for CLIPcap and VLP. Right: Standard METEOR Score ($\uparrow$) for CLIPcap and VLP.

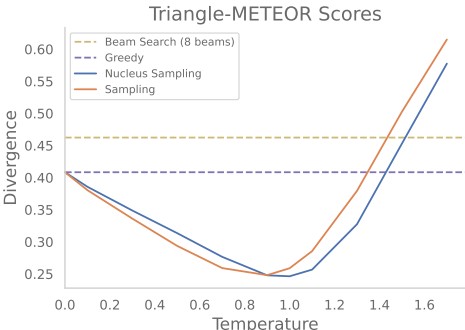 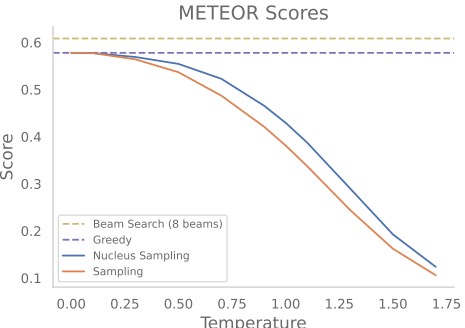

Figure 6: Plots indicating the impact of search technique on divergences. Left: $\text{TRM}_{\text{METEOR}}$ ($\downarrow$) for TVT on MSR-VTT. Right: METEOR Score ($\uparrow$). See appendix A.8 for experimental details.

increase temperature, performance monotonically decreases, giving little insight. Figure 5 illustrates that $\text{TRM}_{\text{METEOR}}$ captures a subtlety in the model comparisons that METEOR does not capture alone: while VLP produces better descriptions at low temperatures, it becomes less fluent (likelihood) on average as we introduce diversity, leading to a change in position. The result is also visible in qualitative samples, given in Figure 4, where we see TRM metrics prioritize both diversity and likelihood. These results confirm observations made by Zhang et al. (2021) for open-ended language generation tasks such as storytelling and dialogue: a fair comparison of approaches must not only compare at the same level of entropy but at a range of entropy levels.

**Sampling algorithms matter**   Not only does the temperature of the generation process matter when correctly trading off between diversity and description correctness (as seen in the previous discussion), but the sampling process itself matters. Figure 6 shows the performance at different temperatures of the Nucleus sampling method (Holtzman et al., 2019) vs. standard sampling, beam search, and greedy, approaches. While maximum-likelihood methods achieve the best METEOR scores, they have relatively high divergence, as they sample only a single description. From Figure 6 we can further see that $\text{TRM}_{\text{METEOR}}$ illustrates how Nucleus sampling allows models to achieve higher temperatures than standard sampling without diverging significantly from the distribution. METEOR alone does not indicate such an effect and only monotonically decreases.

## 5   DISCUSSION AND LIMITATIONS

**Kernel-Based Metrics (KBMs) vs. Triangle-Rank Metrics (TRMs)**   A natural question to ask is: "which metric should practitioners choose when evaluating conditional language models?" KBMs have one major, distinct, advantage over the TRMs in that they are naturally differentiable, yet KBMs also have downsides. The first is that, unlike the TRMs, they require both a pre-trained BERT model and a kernel-density estimator which both have complex behavior affecting the performance of the model. The TRMs, however, can be specified on top of existing natural language distance functions, improving the ability of the user to intuit the model performance. Additionally, TRMs

Table 2: Method evaluation efficiency on the MS-COCO dataset with 5 references and 10 candidates using an intel i7-6850K CPU and 2 NVIDIA Titan X GPUs.

|  | METEOR | TRM$_{METEOR}$ | CIDEr | TRM$_{CIDEr}$ | MMD-BERT | FID-BERT | MAUVE |
|---|---|---|---|---|---|---|---|
| **Samples/Sec** | $298.4 \pm 18.3$ | $161.18 \pm 21.2$ | $131.23 \pm 12.6$ | $97.54 \pm 9.1$ | $53.76 \pm 38.7$ | $17.45 \pm 4.6$ | $2.29 \pm 0.78$ |
| **Wall Time (Min)** | 2.26 | 4.18 | 5.14 | 6.92 | 12.55 | 38.68 | 294.78 |

are bounded and have p-values that can be computed analytically. Finally, because the TRMs do not need a density estimate, they can be more sensitive with small sample sizes (see Figure 3), which is essential for conditional language generation where we have only a few gold-standard samples. Table 2 demonstrates another key benefit of TRMs: efficiency. The time per sample to compute TRMs, while higher than single metric standards, is lower than KBMs on average. In the case of Mauve, computing the p-values is largely intractable (See appendix B.5).

**Perplexity** We acknowledge that perplexity (likelihood of the test distribution) is another alternative metric to proposed methods. While methods for conditional language generation **should** report the perplexity of their models, it has also been noted by Theis et al. (2015) that perplexity can suffer from several issues in the evaluation of generative models. For example, a lookup table storing sufficiently many training examples will produce convincing results but have poor perplexity on the test data. On the other hand, van den Oord & Dambre (2015) demonstrates that even in situations where the perplexity is high, models may not generate high-quality test samples.

**A Discussion on Human Correlation** In this work, we do not provide any experiments correlating our metric scores with human judgments. The reason is two-fold. First, humans are relatively poor at measuring the semantic distance between two distributions in the presence of distractors (Durga, 1980). Further, some evidence has shown that existing decoding methods optimize for fooling humans over correctness (Ippolito et al., 2019). Automated metrics represent a key method for bridging this gap: methods such as TRMs allow us to overcome any deficiencies in human judgment and sensitivity. The second is that the TRM-metrics build on existing, well-established measures with strong human correlation (Vedantam et al., 2015; Lin, 2004; Agarwal & Lavie, 2008; Papineni et al., 2002). Because we do not alter the distance manifold, our contributions only serve to increase the sensitivity of existing measures of semantic similarity, rather than replace the measures altogether. While sensitivity can be enhanced by hacking the p-values of the metrics (such as taking the power of an existing measure), Table 3 demonstrates that low sensitivity is still observed in human leave-one-out experiments (which would not be the case if sensitivity was increased across the board).

**A Note on Reference-Free Metrics** Some metrics, such as CLIP-score (Hessel et al., 2021) for visual description, are immune to ground truth aggregation effects as they are computed in a reference-free way, and focus on pre-trained models' ability to ground vision and language information. Unfortunately, such large, black-box, models represent a liability as a metric as their capabilities are largely unknown, and untested (Floridi & Chiriatti, 2020; Caglayan et al., 2020). Further, the metric is only as good as the model, and CLIP has been known to suffer from numerous issues including counting, attribute-association, and spatial reasoning (Blattmann et al., 2022; Ramesh et al., 2022).

## 6 CONCLUSION

In this work, we have introduced a robust framework for multi-candidate evaluation of conditional language generation models, shown that existing metrics for semantic similarity can be seamlessly extended to this framework, and demonstrated through a case study of visual description that multi-candidate evaluation paired with more sensitive distribution-aware metrics has the power to provide novel insights into existing models and methods. Our method is not without limitations, for example, one of the core drawbacks is the availability of multi-reference data. Outside the field of visual description, it is often not a standard practice to collect more than one gold-standard reference (even in fields such as summarization, where it makes sense to do so). It is necessary for future work to explore how a wider range of existing generation techniques and models perform under this new paradigm, and, as data availability expands, to understand the implications of distribution-aware evaluation in fields beyond visual description. We hope that this step toward a more robust evaluation paradigm will inspire further research into this area of evaluating conditional language generation.

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

# A ADDITIONAL EXPERIMENTAL DETAILS

In this section, we discuss additional experimental details for interested readers.

## A.1 CODE

We make all code/data publicly available for use at `https://s3.us-west-1.wasabisys.com/anon-neurips2022/neurips.tar.gz` (Github link in camera ready). We hope that releasing our code, along with the JSON files containing test-set predictions for the models in question will help inspire further research and examination into the evaluation of models for visual description.

## A.2 DATASETS

**MSR-VTT Dataset:** The MSR-VTT dataset (Xu et al., 2016) is a dataset for video description consisting of 10,000 videos, with 20 reference ground truth descriptions for each video. It was collected by downloading 118 videos for each of 257 queries from a popular video sharing website. MSR-VTT contains 41.2 hours of video, with an average clip length lying between 10 to 30 seconds. It has a vocabulary size of 21,913. For more details about the diversity of the language present in the dataset, we refer readers to Chan et al. (2022).

**MS-COCO Dataset:** The MS-COCO dataset (Lin et al., 2014) is a large-scale dataset for image description, object detection and segmentation. MS-COCO contains 328K images, each with 5 ground truth descriptions generated by human AMT workers. For more details about the diversity of the language present in the dataset, we refer readers to Chan et al. (2022). MS-COCO is licensed under a Creative Commons Attribution 4.0 license.

## A.3 MODELS

This paper explores the performance of our metrics over several models: two video captioning models, and two image captioning models.

**TVT** The Two-View Transformer (Chen et al., 2018) is a baseline method for video description, which consists of a transformer encoder/decoder structure. While we did not have access to the original code, we trained our own version of the model on the MSR-VTT dataset (standard splits), leveraging features from Perez-Martin et al. (2021). The model was trained for 300 epochs, with a batch size of 64, model hidden dimension of 512, 4 transformer encoder and decoder layers with 8 heads each, and dropout of 0.5. For optimization, we leveraged the Adam optimizer with a learning rate of $3e^{-4}$ and weight decay of $1e^{-5}$ with exponential learning rate decay with gamma 0.99. This model achieves a $CIDEr$ score of 56.39 on the test dataset. The model was trained using a Titan RTX-8000 GPU over the course of several hours.

**O2NA** O2NA (Liu et al., 2021) is a recent approach for non-auto-regressive generation of video captions. While the method had available code and checkpoints which we used for this experiment, the method is not designed to sample more than one candidate caption at any given time. To adjust the model to sample multiple candidate captions, we made several adjustments. First, the model was modified to sample a length according to a softmax distribution over the length likelihoods (instead of using a greedy choice of length, or beam search over lengths, as proposed in the paper). Second, the model was modified to sample tokens at each non-autoregressive step from a temperature-adjusted softmax distribution instead of greedily sampling tokens. We make our modified code available as a patch to the original repository, in the hopes that other users will continue to build on these alterations.

**CLIPCap** CLIPCap (Mokady et al., 2021) is a recent model for image description based on using the CLIP (Radford et al., 2021a) model for large vision and language pre-training as a feature encoder, and GPT (Brown et al., 2020) as a natural language decoder. CLIPCap code and MS-COCO trained model checkpoints are publicly available from the authors, however we made some alterations to support temperature-based and nucleus sampling. We make our modified code available as a patch to the original repository, in the hopes that other users will continue to build on these alterations. CLIPCap is licensed under the MIT license.

**VLP**  VLP (Zhou et al., 2020) is a unified vision and language pre-training model, designed to perform both image captioning and visual question answering. The model is pre-trained on the Conceptual Captions (Sharma et al., 2018) dataset, and fine-tuned on the MS-COCO captions dataset for image description. The authors make code and pre-trained models publicly available, however we modified the code somewhat to support additional sampling methods. We make our modified code available as a patch to the original repository, in the hopes that other users will continue to build on these alterations. VLP is licensed under the Apache License 2.0.

## A.4  DISTANCE METRICS

In this paper, we explore three base semantic metrics as distance underlying our TRM methods, CIDEr-D (Vedantam et al., 2015), METEOR (Agarwal & Lavie, 2008), and BERT Distance (Zhang* et al., 2020).

**CIDEr-D**  CIDEr-D (Vedantam et al., 2015) is a n-gram-based metric designed for visual description, and based on the idea that common words are less useful in practice than uncommon words. In practice, this takes the form of a cosine similarity between TF-IDF weighted vectors representing the sentences. Because CIDEr-D is a score, and not a distance, we create a distance function: $d(c, r) = 10 - C(c, r)$, which works as CIDEr-D is bounded by 10. Note that because CIDEr-D is 10 if and only if and only if the two sentences are equal, this fulfills the TRM requirements.

**METEOR**  METEOR (Agarwal & Lavie, 2008) is a score which evaluates the semantic distance between two text utterances based on one-to-one matches between tokens in the candidate and reference text. The score first computes an alignment between the reference and candidate, and computes a score based on the quality of the alignment. Because METEOR is a score, and not a distance function, we use the distance $d(c, r) = 1 - M(c, r)$, where $M$ is the METEOR score of the reference. Because METEOR is bounded at 1 if and only if the two utterances are identical, this simple transformation satisfies the requirements of the TRM adjustment. While we could explore other ways of deriving a distance from METEOR, we found that this simple approach was sufficient to demonstrate the performance of our methods.

**BERT Distance**  A recent method for determining the semantic distance between two samples is to leverage a pre-trained BERT embedding model to create a semantic embedding of the text, and computing the cosine distance between the test samples. In our work, we leverage the `MiniLM-L6-v2` model from the sentence-transformers package by Reimers & Gurevych (2019) to embed our descriptions. Because cosine distance is already a distance function, no additional transformation is necessary.

## A.5  P-VALUE COMPUTATIONS

For our experiments, our null hypothesis is that the candidate samples and the ground truth samples are drawn from the same distribution. Because most of the methods do not have an analytical way to compute the p-values (in fact, the TRMs are the only method which has an analytic p-value computation given in Liu & Modarres (2011)), we instead must compute the p-values though sampling. We thus enumerate the value of the statistic across all of the possible candidate/reference partitions given the joint set of candidates and references, and determine the probability of observing the sampled value, or some value more extreme.

The values in Table 1 represent the p-value obtained with a single candidate sentence, and 4 ground truth candidates for MS-COCO, or 19 ground truth candidates for MSR-VTT. We reserve one gorund truth description in both datasets to serve as the "Human" performance description. For TVT, CLIPCap and VLP, we sample the descriptions using beam search with 16 beams. For O2NA, which is a non-autoregressive model, we sample according to the method suggested in the original work (see Liu et al. (2021)). Because there are several thousand videos per dataset, computing all possible combinations across the dataset would be far from tractable. Thus, the p-values were computed on a per-visual-input basis, and then aggregated across videos using the harmonic mean, as suggested by Wilson (2019). Such an aggregation method is valid when the experiments are not independent (which they are not), unlike Fischer's method (Fisher, 1992).

Figure 3 demonstrates the log p-values for the proposed methods across several candidate samples. For MS-COCO, we use all five reference captions, and between one and ten candidate captions

sampled from CLIPCap using Nucleus Sampling (Holtzman et al., 2019) with a temperature of 1.0, top-p of 0.9 and top-k of 20. The caption set is generated once, meaning that the two-candidate set consists of the one-candidate set and one more additional caption. For MSR-VTT, we use 10 reference captions, and between one and seven candidate captions sampled from O2NA as described in appendix A.3 with a temperature of 1.0 for both the length and token samples. We do not go to the full 10 candidate captions for MSR-VTT due to tractability concerns, since adding an additional caption forces twice the number of partitions to be evaluated when computing p-values.

The above experiments were performed on several n2d-standard-32 cloud GCP instances, containing 32vCPUs and 128GB of RAM.

## A.6 FRECHET BERT DISTANCE

The Frechet Inception Distance, originally proposed in Salimans et al. (2016), has often been used for the evaluation of the distance between samples of images generated by GANs. Images are first embedded in a latent space using a pre-trained inception network, and then the Frechet distance between the generated samples and the reference samples is computed. In our work, we replace the images with text, and the inception network with a pre-trained BERT embedding network (Devlin et al., 2018). For a set of candidate samples $(c_1, \ldots, c_n) = C$, a set of reference samples $(r_1, \ldots, r_m) \in R$, and a BERT embedding function $\phi_{\text{BERT}} : C \cup R \to \mathbb{R}^k$, we compute the Frechet BERT Distance as:

$$d^2 = \left\| \frac{1}{n} \sum_{i=1}^{n} \phi_{BERT}(c_i) - \frac{1}{m} \sum_{i=1}^{n} \phi_{BERT}(r_i) \right\| + \text{Tr}\left( C_C + C_R - 2\sqrt{C_C C_R} \right) \quad (5)$$

where $C_C$ and $C_R$ are the covariance matrices of the $C$ and $R$ sets embedded with $\phi_{\text{BERT}}$ respectively.

To get the BERT embedding, we leverage the CLS token of a large pre-trained model, in this case, the MiniLM-L6-v2 model from the sentence-transformers package by Reimers & Gurevych (2019).

The computation of p-values for the Frechet-BERT distance is largely bottle-necked by the slow performance of the sqrtm function, which, because the matrices are not symmetric, has no efficient algorithm for computation. Additionally, unlike the feature computation, this operation must occur for every partition, leading to significantly reduced efficiency compared to the other measures presented in this paper.

## A.7 MMD-BERT

Another common metric in the GAN literature is the computation of a maximum-mean discrepancy between kernel-estimates of the samples introduced by Li et al. (2017). For a set of candidate samples $(c_1, \ldots, c_n) = C$, a set of reference samples $(r_1, \ldots, r_m) \in R$, and a BERT embedding function $\phi_{\text{BERT}} : C \cup R \to \mathbb{R}^k$, we compute the MMD-BERT distance as:

$$
\hat{MMD} = \sum_{i=1}^{N} \sum_{j=1}^{N} K(\phi_{\text{BERT}}(c_i), \phi_{\text{BERT}}(c_j)) +
$$
$$
\sum_{i=1}^{M} \sum_{j=1}^{M} K(\phi_{\text{BERT}}(r_i), \phi_{\text{BERT}}(r_j)) + \sum_{i=1}^{N} \sum_{j=1}^{M} K(\phi_{\text{BERT}}(c_i), \phi_{\text{BERT}}(r_j)) \quad (6)
$$

where $K$ is a kernel function. In our experiments, we use an RBF kernel function with $\sigma$ equal to the median distance pairwise distance divided by two.

## A.8 SEARCH TECHNIQUES

In section 3, Figure 6, we explore the performance of several different search techniques for our two-view transformer model on the MSR-VTT dataset. In this figure, we explore four decoding search techniques: Greedy Search, Beam Search, Temperature-Based Sampling, and Nucleus Sampling. For each method, and for each video in the test set, we sample 10 descriptions. For Greedy Search, we sample 10 repeated sentences. For beam search we sample the top beam search candidate, and repeat this ten times. While we did explore using the top 10 results from a larger beam search, we found that a smaller beam search and repeated values produced better METEOR scores, so we chose to compare against this. Wider beam searches did produce higher TRM$_{\text{METEOR}}$ scores, but because optimizing for METEOR would be the current paradigm, we decided to include that in the referenced figure.

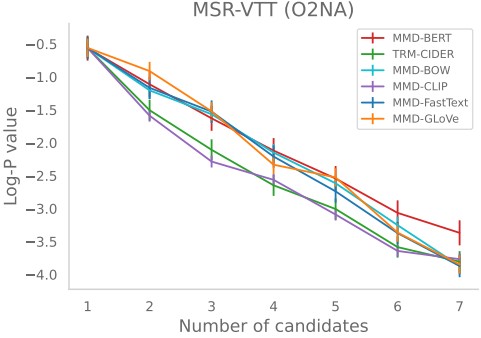

| Sensitivity and performance on Human-Generated Captions | | |
|---|---|---|
| **Method** | **Log-P** | **Samples/Sec** |
| TRM-CIDEr | -1.596 | 88.93 |
| MMD-BERT | -1.786 | 56.68 |
| MMD-CLIP | -1.887 | 14.41 |
| MMD-GLoVe | -1.952 | 54.8 |
| MMD-FastText | -1.954 | 57.45 |
| MMD-BOW | -2.022 | 49.41 |

Figure 7: Performance of several different embedding functions for the MMD-* family of metrics. Left: Sensitivity when evaluated on the MSR-VTT dataset with ten reference captions and between one and seven candidate captions generated by O2NA. Right: Sensitivity and speed when evaluated on human reference samples with 5 references and 5 candidates.

For standard temperature based sampling, we sampled 10 results at each temperature. For Nucleus sampling, we sample 10 results at each temperature, however we freeze they hyper-paramters of top-p at 0.9 and top-k at 20, as we found these values to generate the best scores under the standard pairwise metrics. It remains relevant future work to perform a deep-dive into the different generative methods with respect to TRMs, as there are likely many interesting lessons that can be learned.

## B ADDITIONAL RESULTS

In this section we present several additional interesting results to augment those in the main discussion.

### B.1 EMBEDDING METHODS FOR KBMS

In the main work, we primarily explore a BERT-based embedding method for the kernel-based methods. Such an exploration does not preclude the use of other embedding methods, each of which has different trade-offs, when looking at the quality of the resulting metric, what the resulting metric measures, the time required to compute the embedding, and the performance when the reference distribution is limited to small numbers of human samples (such as happens in practice). Figure Figure 7 shows a quick look at several possible choices for embedding methods in the MMD-* family, including Bag of words (with a 5K vocab), GLoVe (Pennington et al., 2014), FastText (Bojanowski et al., 2017), and CLIP (Radford et al., 2021b).

While we can see that some of the methods are more sensitive to deviations in the image distributions, such methods come with additional trade-offs. CLIP-style embeddings are the most sensitive to human versus generated captions with fewer captions created, but are significantly slower to evaluate at test time (almost 4x slower) than MMD-BERT, and also produce a higher p-value when computing the leave-one scores on the human captions (which is less desirable, as the human captions are drawn from the same distribution).

### B.2 UNIQUE VS. CORRECT DESCRIPTIONS

In Figure 8, we explicitly demonstrate how TRMs enable evaluation of both caption diversity and quality. We artificially generate candidates for the MSR-VTT dataset by mixing human-generated exact descriptions with human-generated descriptions from other videos. On one axis we have the number of unique descriptions and on the other axis we have the number of correct (exactly-matching) descriptions. Clearly, unlike METEOR alone, $TRM_{METEOR}$ scores are affected by both correctness and diversity.

Each experiment consisted of 10 candidate captions from the MSR-VTT dataset, and 10 reference captions from the MSR-VTT dataset. We first split the 20 MSR-VTT reference captions into two sets of 10. One set of 10 captions formed the references. To select the candidate captions, we first sampled $k$ unique captions from the remaining reference set (which formed the "correct pool"), and $k$ unique captions from other videos in the dataset at random (forming the "incorrect pool"). We then selected $m$ correct captions, from the correct pool (at random) and $10 - m$ captions from the

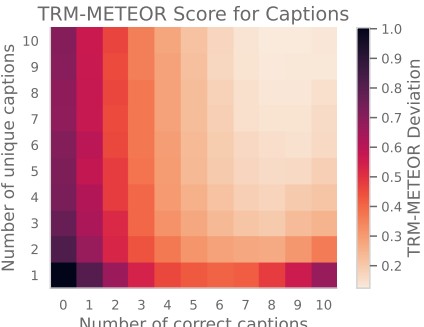 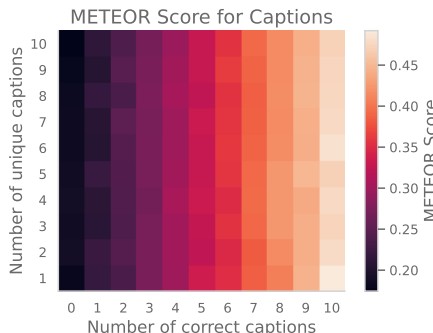

Figure 8: Plots showing how TRMs evaluate both diversity and quality. Left: $\text{TRM}_{\text{METEOR}}$, Right: METEOR. Lighter colors represent better scores. While $\text{TRM}_{\text{METEOR}}$ trades off between diversity and quality, METEOR focuses only on quality not diversity.

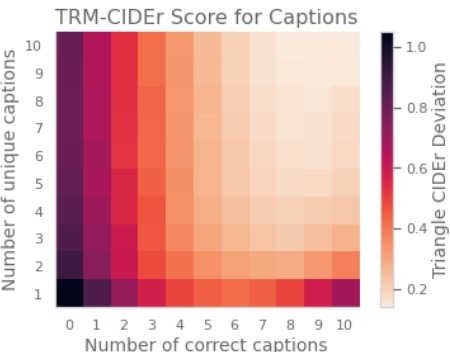 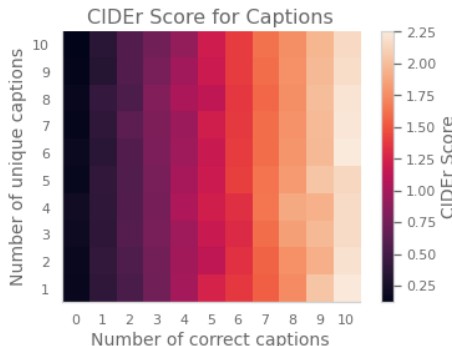

Figure 9: Plots showing diversity vs. quality tradeoffs. Left: $\text{TRM}_{\text{CIDEr}}$, Right: CIDEr. Lighter colors represent better scores. While $\text{TRM}_{\text{CIDEr}}$ trades off between diversity and quality, CIDEr focuses only on quality not diversity.

incorrect pool (at random). This was then plotted with $m$ on the x-axis, and $k$ on the y-axis, as a heat-map, where lighter colors represent better scores (higher METEOR, or lower TRM-METEOR), and darker colors represent poor scores.

We also explored the performance of the CIDEr metric across the same axes, the results of which are shown in Figure 9. We can see that they are largely similar to those from the METEOR metric, suggesting that regardless of the underlying metric, we are still making similar trade-offs between diversity and correctness.

### B.3 Visualizing Central Descriptions

We have found that descriptions which minimize the expected distance to the ground truth distribution are relatively sparse in detail compared to other descriptions. Figures 10, 11, 12 and 13 show qualitative examples of such descriptions for the MS-COCO dataset. Each plot shows qualitative examples of "central" captions. The caption marked with arrows is the ground truth caption which minimizes the expected METEOR distance to the other reference captions, and the other captions are the additional references in the MS-COCO dataset. Images are selected at random, and do not represent cherry-picked samples from MS-COCO.

### B.4 Human p-values

Strong metrics for distributional comparison will have high sensitivity to samples coming from distinct distributions, and will produce high p-values for samples which come from the same distribution. To check that such a relationship holds, we also perform leave-one-out experiments using human-generated captions from the reference set for both MSR-VTT and MS-COCO. For MSR-VTT, we split the reference data into sets of 10 candidate samples and 10 reference samples, and compute the deviations using this partitioning. For MS-COCO, we leverage the c40 split which has 40 reference

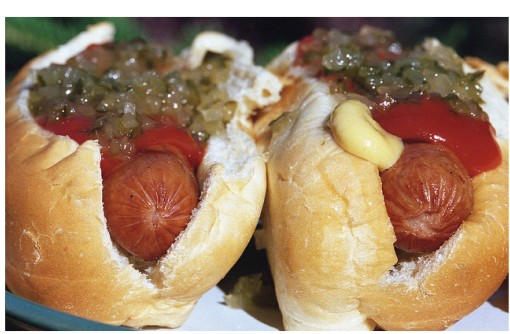

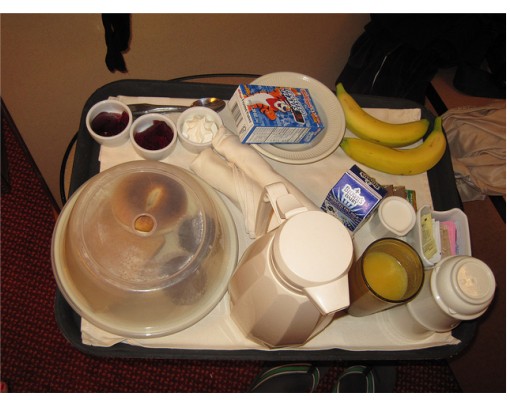

```
Two hot dogs sitting side by side with condiments.
Two hot dogs are laden with relish, ketchup, and mustard.
>>> two hot dogs on a plate loaded with condiments
Two hot dogs covered with ketchup and relish on a plate.
Two hot dogs in buns are smothered with condiments.
```

```
The meal is ready on the tray to be eaten.
A breakfast was delivered to a hotel room on a tray.
a bunch of food and stuff is laying on a tray
>>> Bananas, cereal, juice and other breakfast foods on a tray.
This tray includes several different items for a full breakfast.
```

Figure 10: Qualitative example of "central" captions. The caption marked with arrows is the ground truth caption which minimizes the expected METEOR distance to the other reference captions.

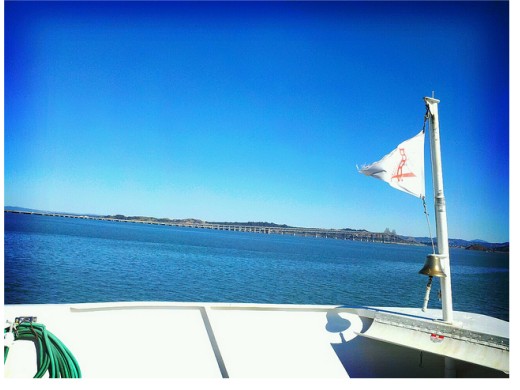

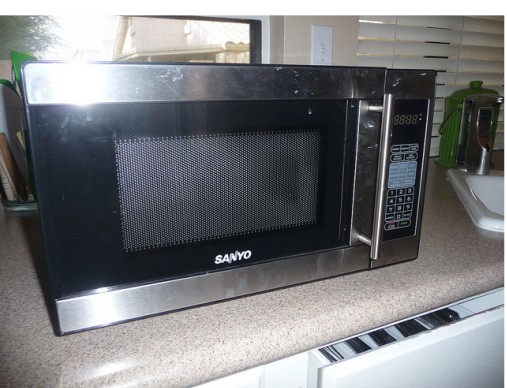

```
A photo taken from a boat with a long bridge in the background.
A view of the coast from within a boat
The side of a boat and a bridge going over the ocean.
>>> A view of the lake, taken from a boat.
A boat flies its flag while sailing just off a pier.
```

```
a microwave on a kitchen counter above a dishwasher
this micro wave is black and silver and is on the counter
>>> A microwave oven sitting on top of a counter.
A microwave sitting on a counter, its stainless steel.
a silver microwave oven on a tan counter and a window
```

Figure 11: Qualitative example of "central" captions. The caption marked with arrows is the ground truth caption which minimizes the expected METEOR distance to the other reference captions.

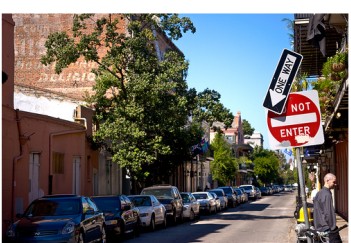

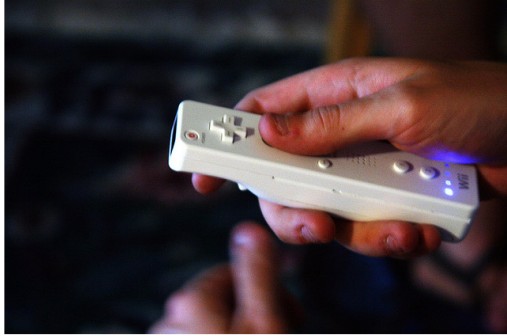

```
A narrow city street has a leaning one way sign.
>>> a street with a line of cars parked on the side
Cars are parked alongside the road and a man is standing next to a sign.
A man is standing next to a road sign with a line of parked cars across the street in an urban area
A crooked one way sign pointing into the ground
```

```
A person pressing a button on a Wii controller.
A hand holds a remote that operates a video game.
There are no image to describe on this page..
>>> A person is holding a white Wii control
someone that is holding a wii remote in their hand
```

Figure 12: Qualitative example of "central" captions. The caption marked with arrows is the ground truth caption which minimizes the expected METEOR distance to the other reference captions.

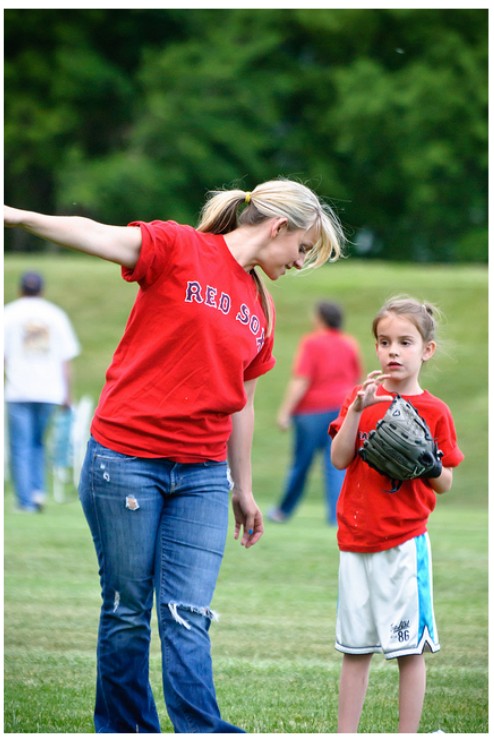

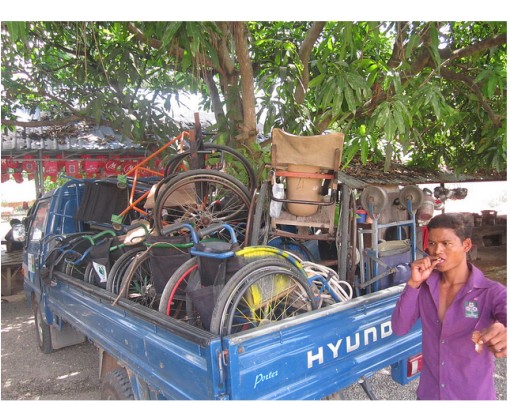

```
a blue truck and a male in a purple shirt and a tree
Blue pickup truck filled with scrap pieces of household items.
A man has filled his truck with wheelchairs.
>>> A blue truck parked next to a tree and a man.
a man standing next to a truck full of bikes and a wheel chair
```

```
Mom gives her daughter a lesson in using her baseball glove.
>>> Mother and her son playing in a few
two girls in red shirts grass and a baseball glove
A woman playing catch with her young child.
The mom is teaching her daughter to play baseball
```

Figure 13: Qualitative example of "central" captions. The caption marked with arrows is the ground truth caption which minimizes the expected METEOR distance to the other reference captions.

Table 3: Log P-Values on human leave-one our samples. We can see that, surprisingly, none of the methods (even the standard aggregations) produce statistically signficant differences. That being said, TRMs often produce higher p-values, indicating that they may be more robust to noise in human caption sets. We do not compute the Frechet-BERT values for humans here, as it was prohibitively expensive.

|  | METEOR | TRM$_{METEOR}$ | CIDEr | TRM$_{CIDEr}$ | BERT | TRM$_{BERT}$ | MMD-BERT |
|---|---|---|---|---|---|---|---|
| **MSCOCO** | -0.6303 | -0.5941 | -0.5957 | -0.4742 | -0.6230 | -0.5633 | -0.6550 |
| **MSR-VTT** | -1.0046 | -0.9613 | -1.0224 | -0.9777 | -1.0172 | -1.040 | -1.0374 |

descriptions for 5000 samples of the ground truth. We partition the references for each video into groups of ten descriptions, and compute the p-values from pairs of these partitions. Table 3 gives the performance of the metrics on this human data.

## B.5 MAUVE PERFORMANCE

In the main work, we found that MAUVE was prohibitively slow to use to compute p-values for the training data. Because our p-values were computed with 10 reference sentences, and up to 10 candidate sentences, at the existing rate, it could take several years to compute the MAUVE p-values for the 50,000 sample MS-COCO dataset. In Table 4, we present several high-variance estimates of the MAUVE p-values (computed using only 100 samples).

Table 4: Log p-value estimates for MAUVE using five candidates, five references, and 100 samples (at nucleus sampling temperature 1.0 for O2NA, CLIPCap and VLP models). We can see that Log p-values for MSR-VTT and MS-COCO are signficantly worse than METEOR even with aggregation, likely due to the method using k-means to approximate the text distributions with only 5 samples.

| Dataset | MAUVE Log p-value | METEOR Log p-value |
|---|---|---|
| MSR-VTT (O2NA) | -0.4414 | -1.7881 |
| MSR-VTT (Human Captions) | -0.1441 | -0.6037 |
| MS-COCO (CLIPCap) | -0.3980 | -2.5585 |
| MS-COCO (VLP) | -0.3234 | -2.8609 |
| MS-COCO (Human Captions) | -0.2189 | -0.7233 |

## B.6 ADDITIONAL QUALITATIVE SAMPLES

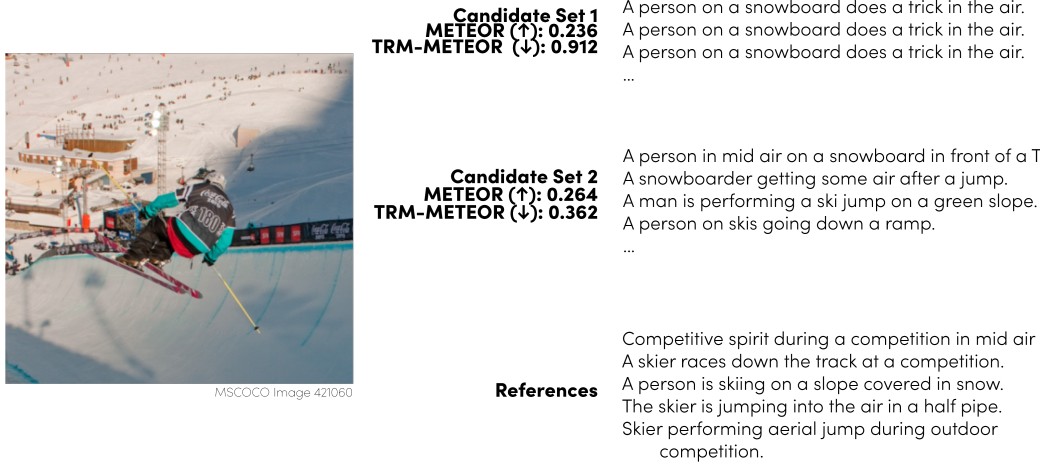

Figure 14: A qualitative sample from CLIPcap. Candidate set one uses beam search (8 beams), while candidate set two uses nucleus sampling (with temperature one, top-k of 20 and top-p of 0.9).

