# OpenReview forum: "Distribution Aware Metrics for Conditional Natural Language Generation"
_ICLR.cc/2023/Conference — Submitted to ICLR 2023_

### Official Review · Reviewer_URmg · 2022-10-18

**Confidence:** 4
**Correctness:** 3
**Technical Novelty And Significance:** 3
**Empirical Novelty And Significance:** Not applicable
**Recommendation:** 5

**Clarity, Quality, Novelty And Reproducibility:**

Clarity, Quality, and Novelty is enough.
Reproducibility of the indicator is high, but experimental reproducibility appears to be not so high due to data dependence.

**Strength And Weaknesses:**

Strength\
*Issues are clear and convincing:  An issue in the evaluation of conditional text generation models from multiple sampled texts.\
*Proposals for new metrics are important and challenging:  Reuse existing text semantic distance functions and increase the sensitivity of these measures when faced with multiple candidates and references, rather than rebuild the entire set of distributional measures.\
*Prior research is sufficient: Discuss the advantage and disadvantages of existing metrics and extend the existing pairwise metrics.

Weaknesses\
*Term definition: For example, many distribution  appear to appear many times in different ways (density or not) many times (e.g., Figure1 and KERNEL-BASED METRICS), but how is this term defined in this paper?\
*Leaning a bit too empirically: Sensitivity to data, reference text, especially the ground-truth and task, is likely to change the evaluation of the indicator, although the Table 3 and its discussion.
Could you apply this framework to other tasks such summarization or information extraction?\
*No human judgments: While metrics such as ROUGE have been pointed out for their proximity to human judgment, the reason why there is no human judgment in the proposed metrics based on traditional metrics is weak.
Can you explain the case study?

**Summary Of The Paper:**

This paper points out problems with traditional metrics for evaluating conditional natural language generation,
and proposes a novel paradigm for multi-candidate evaluation,
as they are not appropriate for domains such as visual description or summarization where are semantically diverse.
In this work, authors introduce metrics to measure deviations between samples from candidate and reference distributions.
They are designed to leverage already strong pairwise tools and alleviate the sensitivity issues induced by introducing simple augmentations to existing pairwise metrics.
Authors demonstrate that existing metrics for semantic similarity can be seamlessly extended to this framework and their paradigm can capture nuances through a case study of visual description.


**Summary Of The Review:**

This paper proposes a novel paradigm for multi-candidate evaluation of conditional language generation models.
It is designed to leverage already strong pairwise tools and alleviate the sensitivity issues induced by introducing simple augmentations to existing pairwise metrics.
Authors propose both Kernel-Based Metrics (KBMs) and Triangle-Rank Metrics (TRMs) as the basis framework,
compare them in their benefits,
and that existing metrics for semantic similarity can be seamlessly extended to this framework.

---

> ### Author Response · Authors · 2022-11-11
> **Reviewer Response**
>
> Thanks for the detailed evaluation of the work! We hope that we can address some of the comments here!
>
> **Term definition (distribution)** - We will work to clarify the word “distribution” in the paper, however, in all cases, we take it to mean the conditional distribution p(t|C), the probability of the generated text given the context. When we refer to the reference distribution, we refer to the ground truth distribution p(t|C), often implied by the reference samples sampled from this distribution. For TRMs and in the most general sense that we use “distribution” in the paper, these distributions are defined by probability mass functions supported over the set of possible captions, which we will make clear in the paper. Unlike TRMs, KBMs work by projecting the discrete captions into a continuous space, making a kernelized estimate of the density function, and directly comparing these density functions using common continuous density divergence measures. We will clarify this in the section on KBMs.
>
> **Empirical Evaluation** - We agree that our experiments measuring the sensitivity of the measure are prone to data/ground-truth variance, and that the measure may have different impacts on different fields (such as summarization, or information extraction). TRMs are well grounded theoretically, as a permutation test applied to the triangle test statistic (Liu & Modarres (2011)), and show strong empirical results on visual description, which we believe is a strong indicator of their value to at least one community (i.e. the visual description community). While we did consider adding additional tasks during the paper writing process, due to the limits of paper length, and to keep the focus of the paper clear, we chose to comprehensively evaluate the performance of visual description tasks instead of performing several cursory evaluations in other domains. Beyond the length of the paper, we were also limited by the availability of ground truth data in other fields. In many other fields (particularly fields such as visual question answering, summarization, and visual storytelling), it is not a norm to collect multiple ground truth captions for each context, and multiple ground truths are a prerequisite for multi-candidate evaluation. We truly hope that this work will inspire others to collect additional data, and evaluate their models in a multi-candidate approach.
>
> **Human Judgements** - We do not include human evaluations in this paper, as this paper does not aim to approximate human judgements of the quality of sets of captions. Unlike pairwise semantic measures which establish a metric space, the proposed algorithm operates within a pre-existing metric space (which could be ROUGE, it could be BERT-embedding vectors, or it could be any other metric space). This paper only serves to measure the distance between empirical distributions (samples from a candidate distribution and samples from a target distribution), which have no explicit predefined distance function. We motivate our work through applications to a field which has traditionally been defined by human judgements of quality, but beyond the discussion in section 5, we believe that there is nothing inherent to our method which requires a human evaluation. Could the reviewer please clarify which "case study" is being referenced? We would be happy to explain further.

---

> > ### Comment · Reviewer_URmg · 2022-11-19
> > **Thanks for the response**
> >
> > Thank you for your detailed response. Some of my concerns are clarified.
> > As the paper points out, if each ground truth itself (reference) is given as multiple texts, generation models tend to yield one of those texts as the target text (candidate), i.e., the reproduction of the ground truth.
> > The rationale for the improvement of these models and its impact on human judgement by this metrics is still unclear.

---

> > > ### Author Response · Authors · 2022-11-23
> > > **Thanks for the discussion!**
> > >
> > > We thank the reviewer for taking time to read and respond to our rebuttal. We seem to be in near-complete agreement. As the reviewer says “if each ground truth itself (reference) is given as multiple texts, generation models tend to yield one of those texts as the target text (candidate)” (emphasis added). This is the core problem which this work addresses, since it follows that model outputs should be evaluated as multiple texts, not just a single text. In contrast to traditional captioning evaluation practices, we are exploring the use of multiple captions to better capture the semantics of the scene. This work provides new distribution aware evaluation methods for this setting, built upon existing metrics which have been shown to correlate with human judgements. Given models which can generate a collection of multiple texts, we can explore further human evaluation of the collections, or distillation of the collections into a single more detailed caption. Both of these are part of future work. We hope this helps explain why it’s important to explore the distribution of the collection of output captions instead of a single output caption.

---

### Official Review · Reviewer_ZEw9 · 2022-10-20

**Confidence:** 3
**Correctness:** 3
**Technical Novelty And Significance:** 2
**Empirical Novelty And Significance:** 3
**Recommendation:** 6

**Clarity, Quality, Novelty And Reproducibility:**

The major problem of the paper is the very laconic intuition provided about its major contribution TRMs. In addition, the paper is in general well written and the discussion of the presented problem and experimental results are adequate. Apart from the proposed TRMs, the novelty of the paper is limited.

**Strength And Weaknesses:**

Strengths:
- The motivation of the paper is interesting, especially for the problem of generating visual descriptions, which is studied in the experimental section.
- Although the novelty of KBMs is incremental, they are appropriate for the problem in-hands.
- The experimental section is appropriate. In particular, the experiment with VLP vs. CLIPCap is particularly enlightening.

Weaknesses:
1. The authors should elaborate more on the intuition of the proposed TRMs (eq (4)). It is clear that a triangle where all edges have the same length will produce a score of 0, but apparently $Q(C,R)$ cannot discriminate between a situation where $I_0 = 1$ and a situation where $I_2 = 1$. Intuitively, a model that can generate a candidate that is closer to the two references than the two references are of each other ($I_0=1$) is better than another model where the candidate is far apart from the references ($I_2=1$). In addition, isn't always $\mathcal{I}(C,R) = |T|$?
2. The authors motivate the need for having multiple candidates by the importance of evaluating diversity in the generated samples. However, it is not clear either how the proposed TRMs take diversity into account.
3. It is unclear what hypotheses testing method is being employed throughout the experimental section to compare if the two distributions are the same. The authors should state clearly which kind of test they have adopted.

Typos:
Eq. (4) - It should be $t_i$ instead of $T_i$ inside the summations.
Section 5, Paragraph about perplexity -" On the other hand, van den Oord & Dambre (2015) demonstrates that even in situations where the perplexity is *low*, models may not generate high-quality test samples."

**Summary Of The Paper:**

The paper argues that current metrics relying on a single candidate text and a single reference are inappropriate to evaluate the quality of conditional language generation models and proposes two families of metrics to evaluate models in a multi-candidate, multi-reference setting: triangle-rank metrics (TRM) and kernel-based metrics (KBM). The experimental results show that both are better than naive aggregation (over multiple candidates) at discriminating between the model and the ground truth distributions.

**Summary Of The Review:**

Although some findings of the paper are interesting, its novel contribution is rather small and poorly explained, so at the moment I feel it is more appropriate to recommend rejection of the paper.

**Update after rebuttal:**
The authors have greatly improved the explanation of the proposed TRM metric and now everything makes much more sense. The proposed method is simple and not technically groundbreaking, but the results show that it can be effective. Its applicability is limited to settings where multiple references are available, though. For all these reasons, I decided to increase my score slightly.

---

> ### Author Response · Authors · 2022-11-11
> **Reviewer Response**
>
> Thanks for the time that you took to review the paper, and looking deeply into the mathematics! We hope that we can address some of your concerns with the following:
>
> **Cannot distinguish between I0/I2** - This is true, our test statistic does not distinguish between these two situations, however, it is not always the case that a model generating a candidate closer to the two references, than the references are to each other, is always a desirable situation (in fact, this is often the situation that we wish to avoid). Consider the situation where the “mean” of all reference captions is generated by the candidate set. This caption is closer to any individual caption than any reference caption may be to other reference captions, however as seen in Figure 1 (and discussed further in references Caglayan et al. (2020), Yeh et al. (2021), and Chan et al. (2022)), such captions capture only the mutual information of the references, and fail to match any particular caption well. We can elaborate on this intuition in the paper - particularly in section 3.2 (given that we can save some space from the simplification provided below).
>
> **I(C, R) = |T|** - Yes! Thanks for pointing out this simplification, we’ll update the paper in the next revision.
>
> **Multiple Candidates** - Our goal is not directly to increase the diversity (or spread) of the captions, but rather, to match the diversity of the captions to the diversity of the references in the semantic space. For TRMs, if the spread of the reference captions does not match the spread of the candidate captions, then there will be an increase in the test statistic. If the spread of the reference captions is low, and the model produces candidate captions with high spread, then the within-sample edges of candidate triangles will be large, while within-sample edges of reference candidates will always be small, leading to high divergence. If the spread of the reference captions is high, and the spread of the candidate captions is small, then the within-sample edges of the candidates will be small, and the within-sample edges of the references will be high, leading to high TRM divergence. This can be seen empirically in Figure 4, where because the spread of candidate set 1 does not match with the spread of the reference set, the TRM-METEOR divergence is high (even though METEOR itself is high), while in candidate set 2, the TRM-METEOR divergence is low (even though METEOR itself is low). We will elaborate further in the next revision of the paper.
>
> **Unclear Hypothesis Testing Method** - In this work we use a permutation test [1], which, while very expensive, allows us to obtain p-values for all of our test-statistics (most of which have entirely unknown sample distributions). We will make this clear in the paper.
>
> [1] Fisher, Ronald Aylmer. "The design of experiments." The design of experiments. 2nd Ed (1937).

---

> > ### Comment · Reviewer_ZEw9 · 2022-11-19
> > **Thank you for the response**
> >
> > Dear authors,
> >
> > Thank you for your response, which has improved my opinion about your work. I have updated my review and increased my score slightly.

---

> > > ### Author Response · Authors · 2022-11-23
> > > **Thanks for the discussion**
> > >
> > > Thanks for continuing the discussion! We’re happy to continue to clarify any other parts of the work for which you still have questions.

---

### Official Review · Reviewer_niHx · 2022-10-29

**Confidence:** 2
**Correctness:** 3
**Technical Novelty And Significance:** 3
**Empirical Novelty And Significance:** 3
**Recommendation:** 6

**Clarity, Quality, Novelty And Reproducibility:**

Originality is good, but I still have some questions about the experiment settings and results.


**Strength And Weaknesses:**

The main strength of this paper is suggesting an interesting idea to evaluate generated texts, especially measuring the diversity of candidates and the quality of them together. I like the idea - the triangle-rank approach since it is a meta-approach that can use other evaluation metrics. And the idea is also simple, so other researchers can adopt and extend the idea for their own research work.

What concerns me most is the lack of experiments on human evaluation. Yes, the authors mentioned that comparing with human evaluations is hard and I agree with that. Human hard to figure out the diversity among items, especially sentences. Even though it is hard to do experiments, it would be better to show the human correlation since this paper suggests new evaluation methods. One way is inserting a wrong candidate with well-generated candidates and asking what the wrong sentence is to humans and suggested methods. Human annotators find it hard to answer the score of the diversity of items, but they easily figure out the weird item [1, 2]. I assume that the authors did a similar experiment in Table 3, but I am not sure.

And I have more questions about this paper. I hope to listen to the author's response and discuss how to improve the paper.

- I am not sure if I understand the meaning of the results (p-value) correctly. The p-value is the probability of chance to see the test statistics if the null hypothesis is true. So, intuitively, we can reject the null hypothesis with a low p-value (i.e., 0.01). Here, the null hypothesis is that samples from the candidate set and reference set are drawn from the same distribution. Then, using BLEU@4 on the MSR-VTT dataset (Table 1), we can say that one ground-truth sentence (Human) has a higher probability that supports the null hypothesis than sentences from machine learning models (TVT and O2NA).
- The authors insist that existing single-ground truth comparison is not sensitive enough. That’s because existing metrics are not good? Or is it the limitation of single-ground truth comparison? If the first assumption is correct, what are the ideal values in Table 1?
- Can the suggested method consider the recall in terms of reference samples? In Figure 1, there are no model samples for left-bottom ground truth reference. Then, the recall value is low because of the non-existence.
- What is BERT in Table 1? BERTScore?

[1] Chang, Jonathan, Sean Gerrish, Chong Wang, Jordan Boyd-Graber, and David Blei. "Reading tea leaves: How humans interpret topic models." Advances in neural information processing systems 22 (2009).
[2] Li, Margaret, Jason Weston, and Stephen Roller. "Acute-eval: Improved dialogue evaluation with optimized questions and multi-turn comparisons." arXiv preprint arXiv:1909.03087 (2019).


**Summary Of The Paper:**

This paper presents new diversity and quality evaluation mechanisms for natural language generation models, especially multiple candidates and multiple reference items. The main idea of the methods is making triangles; those nodes are sentences in the candidate set, and the reference set and edges are existing semantic similarity methods such as BLEU and ROUGE. And the authors assume that two sentences in the same set are closed rather than another sentence. The authors also suggest kernel metrics by using pre-trained large language models to compute the similarity between sentences like a BERTScore. The authors choose to generate descriptions from visual datasets as a case study since the datasets have multiple references for a given data. To show the performance of metrics, the authors suggest using a p-value that shows the probability of rejecting the null hypothesis - samples from the candidate set and reference set are drawn from the same distribution. Experiments show that suggested methods capture the diversity of candidates when the number of candidates are increased and the temperature is changed by showing the decreasing p-value.

**Summary Of The Review:**

I would like to discuss this paper with the authors to make the final decision.

---

> ### Author Response · Authors · 2022-11-11
> **Reviewer Response**
>
> Thanks for the thoughtful comments, and your detailed evaluation of the paper! We'd like to take a chance to respond to some of the comments here.
>
> **P-Value Meanings** - In this work, we introduce TRMs and KBMs, which are both measures of divergence, and, more importantly, test statistics in the rigorous mathematical sense. Thus, the p-values correspond directly to the probability of rejecting the null hypothesis. In our work, the null hypothesis is that *the candidate and reference samples come from the same caption distribution* (which we will make clear in a revision). Thus, the intuition that “we can say that one ground-truth sentence (Human) has a higher probability that supports the null hypothesis” is correct, but because the p-values are so high, it is difficult to draw certain conclusions, since a p-value of 0.65 means that there is a 65% chance that if the experiment was re-run, the divergence observed would be at least as extreme as the measured divergence in the experiment. This means that some measures, such as CIDEr are more sensitive, but BLEU@4, for example, requires many trials to draw conclusions about the degree of divergence of the distributions. For conditional natural language evaluation, we have only so many trials (in many cases, just a single trial), so Table 1 suggests that existing measures cannot really allow us to draw conclusions about how close the model candidate distribution is to the implied distribution of the reference samples. It is worth noting that in this work we make the inherent assumption that the goal of a model should be to produce captions from the same generated distribution as the implied reference distribution.
>
> **Single-ground truth comparison is not sensitive enough** - This appears to be a fundamental limitation of single-caption evaluation. While we cannot say it definitively, from the evidence presented in tables 1 and 3, almost all single-caption measures are not sensitive enough to measure, with significance, if two distributions are the same or different with only a single caption (table 1). The existence of a highly-sensitive single-caption pairwise measure is not impossible, but seems unlikely given our results.
>
> **Considering Sample Recall** - We agree that sample recall would make for an interesting additional measure on top of the TRMs and KBMs which we present in this paper. Both TRMs and KBMs allow us to measure how much the conditional distribution of the text diverges from the reference distribution, which is, itself, a natural form of “recall,” as if the two distributions have different support, then they will have high divergence. Explicitly measuring the recall of semantic concepts for single captions, such as the idea that we could generate a single caption which “recalls” all of the information from the references, or in a more explicit 1:1 fashion is a very interesting direction for future work.
>
> **What is BERT** - Yes, this is the BERTScore - we will make that clear in the next revision.
>
> **Human Experiments** - We do not include human evaluations in this paper, as this paper does not aim to approximate human judgements of the quality of sets of captions. Unlike pairwise semantic measures which *establish a metric space*, the proposed algorithm operates *within a pre-existing metric space* (which could be BLEU, it could be BERT-embedding vectors, or it could be *any other metric space*). This paper only serves to measure the distance between empirical distributions (samples from a candidate distribution and samples from a target distribution), which have no explicit predefined distance function. We motivate our work through applications to a field which has traditionally been defined by human judgements of quality, but beyond the discussion in section 5, we believe that there is nothing inherent to our method which requires a human evaluation.
>
> In designing human experiments, it’s possible that even using outliers may not be enough to help humans correctly detect distributional shifts. While humans may find it easy to detect outliers, just because a single caption is wrong, doesn’t mean that the distributions have high divergence (in fact, such a distribution is mostly correct, and should produce lower divergence scores than distributions that differ in many subtle ways). Designing experiments in such a way that allows humans to easily measure the distance between sample distributions is a challenging, and we believe, highly underexplored, area of human-studies research, which, while potentially interesting, is largely beyond the scope of this work.

---

### Decision · Program_Chairs · 2023-01-20

**Decision:**

Reject

**Justification For Why Not Higher Score:**

Defining an evaluation metric for text generation without running a human evaluation on whether the metric prefers groups of sequences with greater diversity and quality does not validate whether the metric works in practice

**Justification For Why Not Lower Score:**

N/A

**Metareview: Summary, Strengths And Weaknesses:**

The paper argues that current metrics relying on a single candidate text and a single reference are inappropriate to evaluate the quality of conditional language generation models. The authors propose a novel paradigm for multi-candidate evaluation (specifically for tasks such as visual description or summarization where generations may be semantically diverse). In this work, authors introduce metrics to measure deviations between samples from candidate and reference distributions.

**Strengths:**
Overall, the main strength of this paper is the proposal of a novel paradigm for evaluating conditional natural language generation models and a new family of metrics that compare the distributions of reference and model-generated text sets.

**Weaknesses:**
There are some concerns about the lack of human evaluation and the lack of clear explanation on how the proposed metrics take diversity into account. The authors have done a good job of resolving the second concern, but I think this paper would be stronger if a human evaluation was attempted. The same aspects of human evaluation would be true of Mauve (Pillutla et al., 2021), and yet the MAUVE work did explore the correlation of MAUVE with human judgments.


**Summary Of Ac-Reviewer Meeting:**

N/A